# Rethinking Fine-tuning Through Geometric Perspective

**Krishna Sri Ipsit Mantri**
Department of Computer Science
Purdue University
West Lafayette, IN
`mantrik@purdue.edu`

**Moshe Eliasof**
Department of Applied Mathematics
University of Cambridge
Cambridge, United Kingdom
`me532@cam.ac.uk`

**Carola-Bibiane Schönlieb**
Department of Applied Mathematics
University of Cambridge
Cambridge, United Kingdom
`cbs31@cam.ac.uk`

**Bruno Ribeiro**
Department of Computer Science
Purdue University
West Lafayette, IN
`ribeirob@purdue.edu`

## Abstract

Fine-tuning pre-trained neural networks has become a cornerstone of transfer learning. However, the practical success of existing methods like low-rank adaptation (LoRA) lacks theoretical explanation. We introduce geometry-guided fine-tuning, a novel paradigm that models the fine-tuning process as the subtle movement of pre-trained weights on a low-dimensional manifold. Our approach formalizes this process through a learnable ordinary differential equation (ODE) - based framework that controls the search space of the weights, bridging existing methods with geometric principles. We empirically evaluate our method in the context of multi-task learning (MTL) fine-tuning of hierarchical vision transformers in computer vision. We propose a parameter-efficient ODE and evaluate it on the PASCAL-Context MTL benchmark. Our approach, dubbed DELORA offers competitive performance across multiple dense prediction tasks, reducing trainable parameters by up to $4\times$ compared to the best-performing baseline. This work advances both the theoretical understanding and practical application of fine-tuning, promoting efficient learning in resource-constrained environments.

## 1 Introduction

The success of large pre-trained neural networks across various domains, such as GPT-2 [Radford et al., 2019] and Stable Diffusion [Rombach et al., 2022], has made fine-tuning an essential technique in transfer learning. As these models grow in size and complexity, efficient and effective fine-tuning methods become increasingly crucial. While approaches like Low-Rank Adaptation (LoRA) [Hu et al., 2022] have demonstrated practical success, they lack a geometric foundation that guides their design philosophy and accounts for their practical effectiveness.

Several works study the relationship between differential equations and neural networks, covering areas such as architecture design [Chen et al., 2018], stability [Haber and Ruthotto, 2017], and activation function design [Mantri et al., 2024, Chelly et al., 2024]. Taking a different approach, this paper proposes using an ordinary differential equation (ODE) to adapt learned neural network weights through a velocity field.

We introduce a novel fine-tuning paradigm: *geometry-informed fine-tuning of neural networks for multi-task learning (MTL)*. During training, the weights of over-parameterized neural networks often converge to a low-dimensional hyperplane, with an intrinsic dimension much smaller than the nominal parameter count [Li et al., 2018, Aghajanyan et al., 2020]. Inspired by this, we hypothesize that the weights of large pre-trained models, such as hierarchical vision transformers [Liu et al., 2021], lie on a low-dimensional manifold within the high-dimensional parameter space. Fine-tuning can then be modeled as the process of moving these weights on this manifold while optimizing performance on downstream tasks.

This geometric perspective on fine-tuning offers several potential advantages:

1. It provides a generalized and geometrically grounded approach to fine-tuning, potentially situating existing low-rank adaptation methods, like LoRA [Hu et al., 2022], within a broader theoretical framework.

2. It allows control of inductive biases through the choice of geometric structures and dynamics in the fine-tuning process.

3. It enables the use of differential geometry tools to analyze the behavior of fine-tuned models and gain insights into the weight spaces of large pre-trained models.

4. It opens new avenues for developing more efficient and interpretable transfer learning techniques, potentially leading to significant reductions in computational resources required for adaptation to new tasks.

Our framework introduces a velocity-field-based dynamics approach to formalize the transportation of weights on their manifold during fine-tuning for MTL. This novel perspective bridges the gap between the empirical success of parameter-efficient fine-tuning methods and theoretical understanding, offering both insights into existing techniques and directions for developing new, principled approaches.

## 2 Geometry-Guided Fine-tuning

In this section, we present a conceptual framework for geometry-guided fine-tuning of neural networks, grounded in principles from differential geometry and dynamical systems. Within the context of MTL, we show in Section 2.2 how an ODE can be used to develop a broader theoretical framework. In Section 2.3, we then demonstrate a theoretical connection between our framework and a widely-used parameter-efficient fine-tuning method, LoRA [Hu et al., 2022].

### 2.1 Background

We begin by formally introducing Multi-Task Learning (MTL). Let $\mathcal{T} = \{T_1, \cdots, T_K\}$ represent a set of $K$ tasks. We consider a large model $F_{\boldsymbol{\theta}} : \mathcal{X} \to \mathcal{Y}$ with pre-trained parameters $\boldsymbol{\theta} \in \mathbb{R}^n$, where $\mathcal{X}$ denotes the input space and $\mathcal{Y}$ denotes the output space. For each task $T_k$, we have a task-specific dataset $\mathcal{D}_k = \{(x_i, y_i^k)\}_{i=1}^{N_k}$ where $x_i \in \mathcal{X}$ and $y_i^k \in \mathcal{Y}_k$ are the input-output pairs for task $k$, with a shared input space across all tasks. The MTL objective is formulated as

$$\min_{\boldsymbol{\theta}} \sum_{k=1}^{K} \lambda_k \, \mathcal{L}_k(\boldsymbol{\theta}; \mathcal{D}_k) \tag{1}$$

where $\mathcal{L}_k$ denotes the task-specific loss function, and $\lambda_k$ are task-specific weighting factors.

### 2.2 Proposed Method

We hypothesize that the weights of a pre-trained neural network lie on a low-dimensional manifold $\mathcal{M}$ within the high-dimensional parameter space. This manifold encapsulates the learned features and structure from the pre-training process [Mao et al., 2024]. Formally, we posit the existence of a smooth manifold $\mathcal{M} \subset \mathbb{R}^n$ of dimension $d \ll n$ such that $\boldsymbol{\theta} \in \mathcal{M}$. The question then becomes: **Can we efficiently fine-tune large pre-trained models by guiding the MTL objective along this manifold $\mathcal{M}$?**

To achieve this, we constraint the movement of neural network weights $\boldsymbol{\theta}$ to $\mathcal{M}$ during fine-tuning using the flow of a learned smooth velocity field $v^\phi$ with parameters $\phi$. By learning task-specific velocity fields through a neural network, we adapt the pre-trained model to the set of MTL tasks $\mathcal{T}$. Formally,

**Definition 2.1** (Velocity Field for Fine-Tuning). Let $v^\phi : \mathcal{M} \times \mathbb{R} \to T\mathcal{M}$ be a smooth vector field on $\mathcal{M}$, where $T\mathcal{M}$ is the tangent bundle of $\mathcal{M}$. The evolution of the weights during fine-tuning can be described by the flow of $v^\phi$ given by the ODE:

$$\frac{d\boldsymbol{\theta}}{dt} = v^\phi(\boldsymbol{\theta}, t), \tag{2}$$

with the initial condition $\boldsymbol{\theta}(0) = \boldsymbol{\theta}$, where $\boldsymbol{\theta}$ are the pre-trained weights of the network.

The gradient descent step for geometry-guided fine-tuning can be represented using the forward Euler discretization as follows:

**Definition 2.2** (Geometry-Guided Fine-Tuning). For a smooth velocity field $v^\phi$, the parameters $\phi$ and $\boldsymbol{\theta}$ are updated during fine-tuning as follows:

$$\boldsymbol{\theta}_{p+1} = \boldsymbol{\theta}_p + \underbrace{v^\phi(\boldsymbol{\theta}_p, t_p)}_{\text{1-step Forward Euler of Equation (2)}} - \eta \sum_{k=1}^{K} \lambda_k \nabla_{\boldsymbol{\theta}}(\mathcal{L}_k(\boldsymbol{\theta}_p, \phi_p; \mathcal{D}_k)) \tag{3}$$

$$\phi_{p+1} = \phi_p - \eta \sum_{k=1}^{K} \lambda_k \nabla_{\phi}(\mathcal{L}_k(\boldsymbol{\theta}_p, \phi_p; \mathcal{D}_k)) \tag{4}$$

where $p$ is the iteration of gradient descent, and $\eta$ is the learning rate.

## 2.3 Connection to LoRA

We now show how Low-Rank Adaptation (LoRA) [Hu et al., 2022] can be modeled within our geometric framework, threby providing a theoretical grounding for its empirical success. In LoRA, weight updates are constrained to a fixed low-rank subspace. Using the notation from [Hu et al., 2022], the weight update of LoRA excluding the gradients of the loss function is given by

$$\mathbf{W} = \mathbf{W}_0 + \mathbf{B}\,\mathbf{A} \tag{5}$$

where $\mathbf{W}_0 \in \mathbb{R}^{d \times r}$ is the pre-trained weight matrix (frozen), $\mathbf{B} \in \mathbb{R}^{d \times r}, \mathbf{A} \in \mathbb{R}^{r \times k}$ are learnable small rank matrices with $r \ll \min(d, k)$ and $\mathbf{W}$ is the fine-tuned weight matrix. Equation (5) is equivalent to one-step discretization of an ODE with a constant (in time) velocity field $v^\phi = \mathbf{B}\,\mathbf{A}$ where $\phi$ are the entries of $\mathbf{B}$ and $\mathbf{A}$. That is

$$\frac{d\mathbf{W}}{dt} = \mathbf{B}\,\mathbf{A} \tag{6}$$

with initial condition $\mathbf{W}(0) = \mathbf{W}_0$. This formulation not only provides a geometric interpretation of LoRA but also suggests ways to generalize and improve upon it by considering more complex manifold structures or velocity fields. Moreover, it demonstrates how our framework can recover existing methods under specific choices of velocity fields and discretization techniques, while opening avenues for developing new, theoretically grounded fine-tuning techniques.

## 3 Experiments

In this section, we present implementation details of DELORA using the Swin-Tiny backbone [Liu et al., 2021] and conduct preliminary experiments on multi-task learning for dense prediction tasks, including human part segmentation, semantic segmentation, surface normal estimation, and saliency detection.

## 3.1 DELORA

We implement our method using PyTorch and conduct all experiments on a single NVIDIA A100 GPU. Following the approach used in LoRA [Hu et al., 2022], we restrict our method to only the

attention weight matrices $\mathbf{W}_q, \mathbf{W}_v$, which are typically present in the Transformer layer [Vaswani et al., 2017]. For a weight matrix $\mathbf{W} \in \mathbb{R}^{d \times k}$, we compute a parameter-efficient velocity field as follows:

$$v(\mathbf{W}, t) = \text{softmax}\left(\frac{(\mathbf{W}_2 \mathbf{W})(\mathbf{W} \mathbf{W}_1)^\top}{\sqrt{d}}\right) \tag{7}$$

where $\mathbf{W}_1 \in \mathbb{R}^{k \times r}$ and $\mathbf{W}_2 \in \mathbb{R}^{r \times d}$ are learnable projection matrices, $r \in \mathbb{N}$ is a hyperparameter that determines the projection dimension, typically chosen such that $r \ll \min(d, k)$. Given the combination of an ODE perspective and the low-rank parameterization of the learned velocity field, we refer to our method as DELORA.

## 3.2 Results

**Setup.** We evaluate our proposed geometry-guided fine-tuning approach on the PASCAL-MTL benchmark [Everingham et al., 2010], following the PASCAL-Context split used in MTLoRA [Agiza et al., 2024]. This dataset, widely used for multi-task learning (MTL) in dense prediction tasks, includes annotations for semantic segmentation, human part segmentation, surface normal estimation, and saliency detection. The PASCAL-Context dataset consists of 4,998 images in the training split and 5,105 images in the validation split. We apply the same data preprocessing and augmentation techniques as described in MTLoRA [Agiza et al., 2024] to ensure a fair comparison.

**Model Architecture.** Following MTLoRA [Agiza et al., 2024], we implement our approach using a Swin-Tiny transformer backbone [Liu et al., 2021], pre-trained on the *Imagenet22k* [Deng et al., 2009] dataset. The model follows an MTL architecture with a shared encoder and task-specific decoder for each of the four tasks.

Table 1: Comparison with SOTA parameter efficient fine-tuning methods. The table summarizes the number of trainable parameters in each method. The last column indicates whether the model allows simultaneous execution of all tasks.

| Method | SemSeg (mIOU ↑) | Human Parts (mIOU ↑) | Saliency (mIOU ↑) | Normals (rmse ↓) | Trainable Parameters (M) (M) | Single Inference For All Tasks |
|---|---|---|---|---|---|---|
| Single Task | 67.21 | 61.93 | 62.35 | 17.97 | 112.62 | × |
| MTL - Full Fine Tuning | 67.56 | 60.24 | 65.21 | 16.64 | 30.06 | ✓ |
| LoRA [Hu et al., 2022] | 70.12 | 57.73 | 61.90 | 18.96 | 2.87 | × |
| MTLoRA [Agiza et al., 2024] | 67.90 | 59.84 | 65.40 | 16.60 | 8.34 | ✓ |
| DELORA (Ours) | 69.72 | 58.22 | 58.84 | 19.23 | **2.42** | ✓ |

**Discussion.** Our results are summarized in Table 1, where we compare DELORA against MTLoRA [Agiza et al., 2024], LoRA [Hu et al., 2022], single task fine-tuning and MTL full fine-tuning approaches. Our DELORA achieves competitive performance across all tasks:

1. **Parameter Efficiency:** Our DELORA achieves competitive performance while using only 2.42M trainable parameters. Although LoRA has a similar number of parameters, our method achieves better overall performance through joint training and enables single inference for all tasks.

2. **Multi-Task Balance:** Our approach shows a better balance across tasks, at times even outperforming MTLoRA, while using $4\times$ fewer trainable parameters. This highlights the benefits of an ODE based approach and the choice of the velocity field.

These preliminary results provide strong evidence for the potential of geometry-guided fine-tuning. They demonstrate that DELORA can achieve parameter-efficient adaptation while maintaining or enhancing performance across multiple tasks.

# 4 Conclusions and Discussion

This paper introduces geometry-guided fine-tuning, a paradigm that frames the adaptation of pre-trained neural networks within the context of differential geometry and dynamical systems, introducing an early variant called DELORA. Our approach constrains fine-tuning by using the flow of a velocity field on the manifold of neural network parameters, offering several key advantages:

1. A broader theoretical framework that encompasses existing methods like LoRA and enables the development of new, principled techniques.

2. Insights into neural network behavior during task adaptation, viewed through the lens of manifold structures and dynamical systems.

3. Promising empirical results demonstrating competitive performance with significantly fewer parameters.

This combination of theoretical grounding and practical efficiency positions geometry-guided fine-tuning as an promising new direction in transfer learning.

**Research Directions.**   Our geometry-guided fine-tuning framework suggests several promising directions for future research:

1. **Weight Manifold Theory:**
   - **Geometric Properties:** Examine the curvature, dimensionality, and connectivity of weight manifolds across architectures [Aghajanyan et al., 2020, Mao et al., 2024], aiming to better understand model capacity and weight configuration landscapes.
   - **Learning Dynamics:** Study how manifold geometry impacts optimization paths, particularly in terms of convergence behavior and efficiency in reaching task-specific weights [Haber and Ruthotto, 2017].

2. **Enhanced Fine-Tuning Dynamics:**
   - **Tangent Space Linearization:** Build on work in tangent space linearization [Ortiz-Jimenez et al., 2024] to enhance weight disentanglement, facilitating task adaptation within our geometry-guided framework.
   - **Advection Techniques:** Integrate advection mechanisms to guide weight updates along structured paths [Zakariaei et al., 2024], potentially improving stability and preserving relevant model properties.
   - **Adaptive Velocity Fields:** Develop strategies for learning velocity fields that adaptively direct weights to task-specific configurations, informed by model performance or data.

3. **Task-Specific Adaptation:**
   - **Input-Adaptive Tuning:** Enable models to dynamically adjust parameters based on input, using geometry-guided velocity fields for more targeted navigation of the weight manifold.
   - **Geometric Regularization:** Apply Riemannian distance [Kirkpatrick et al., 2017] as a regularizer to keep fine-tuned weights close to pre-trained configurations, potentially enhancing generalization.

4. **Theory and Generalization:**
   - **Convergence Analysis:** Investigate conditions for guaranteed convergence and factors influencing convergence rates within geometry-guided tuning.
   - **Generalization Bounds:** Establish generalization bounds for geometry-guided fine-tuning to assess model performance on new data, considering the effects of manifold constraints [Chen et al., 2018].

By pursuing these research directions, we aim to deepen the theoretical understanding of geometry-guided fine-tuning and develop practical techniques that enhance the efficiency and effectiveness of transfer learning in neural networks. Incorporating insights from tangent space linearization and task arithmetic offers a promising pathway to more robust and adaptable models. We encourage the research community to engage with these challenges, which hold significant potential for advancing the field.

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
