# OpenReview forum: "Rethinking Fine-tuning Through Geometric Perspective"
_NeurIPS.cc/2024/Workshop/UniReps — UniReps_

### Official Review · Reviewer_1N5B · 2024-09-28
**Fine-tuning from a geometric perspective**

**Rating:** 6
**Confidence:** 3

**Review:**

The paper proposes to model fine-tuning process as moving weights on a low-dimensional manifold.

Strength: 1. The paper clearly discusss the advantages of the framework. 2. Empirical evidence demonstrates the effectiveness of the proposed method.

Weakness: 1. Possibly due to constraints on paper length, it is not clear to me how does the concept in Definition 2.2 explains LoRA. 2. Regularization using a Riemannian metric is not new in deep learning; e.g. it appears in continual learning [1]. It might be interesting to discuss the connection.

[1] Overcoming catastrophic forgetting in neural networks, Kirkpatrick et al., https://arxiv.org/abs/1612.00796

---

### Official Review · Reviewer_QmEU · 2024-10-03
**The paper proposes analyzing the fine-tuning of large pre-trained models using ODEs on a low-dimensional Riemannian manifold.**

**Rating:** 6
**Confidence:** 4

**Review:**

The paper tackles an interesting and timely problem, which is the fine-tuning of the large pre-trained model. They propose to analyze the fine-tuning process through the lens of dynamical systems (via ODEs). In particular, they define the ODE on a Riemmanin manifold representing a low-dimensional space, where the hypothesis is that the parameter lies in this low-dimensional manifold.
A connection with LoRA is given, and experimental results showcase the advantage of framing fine-tuning in terms of ODEs.
The paper is a good trigger for exploring this idea. Furthermore, it is well-written and clear.

I have some comments/questions for the future:

1. It is difficult to see how (1) and (2) are related. Based on the paper, we can frame the weights' evolution as an ODE, but how are they related to the loss function in (2)? The vector field is the gradient of the loss function when it is defined as (2).

2. In the Section that connects the method with LoRA, how the parameters \theta are related to the low-rank matrices B and A? Is $\theta = BA$? I suggest expanding on this because it is not clear. Looking at the Euler discretization, it seems that BA is the gradient (velocity field), but it isn't easy to follow without clarifying the relationship.

3. How is defined $\mathcal{L}_{task}$ in the experiments? And it says that a regularizer is not used; this means that $ d_g(\theta, \theta_0)$ is 0. Why?

4. How is the practical algorithm of DELoRA compared to LoRA? In DELoRA, the update is given by (3) If this is true, then it is confusing the phrase "Equation (3) is precisely the single-step LoRA update.."

Minor comments:

1. Some important references are missing in the context of fine-tuning pre-trained models; for instance, [1].

2. LoRA citation is the arxiv version instead of the conference paper.
[1] Ortiz-Jimenez, Guillermo, Alessandro Favero, and Pascal Frossard. "Task arithmetic in the tangent space: Improved editing of pre-trained models." Advances in Neural Information Processing Systems36 (2024).


[1] Ortiz-Jimenez, Guillermo, Alessandro Favero, and Pascal Frossard. "Task arithmetic in the tangent space: Improved editing of pre-trained models." Advances in Neural Information Processing Systems 36 (2024).

---

### Official Review · Reviewer_4HWv · 2024-10-05
**The paper introduces a novel geometry-guided fine-tuning method for neural networks with promising theoretical backing and parameter efficiency, but its claim of providing a unified framework is less well-substantiated due to limited empirical validation beyond specific cases like LoRA.**

**Rating:** 6
**Confidence:** 3

**Review:**

## Summary
The paper presents a novel approach to fine-tuning pre-trained neural networks, framed from a geometric perspective. The authors propose geometry-guided fine-tuning, which models the movement of pre-trained weights on a low-dimensional manifold using an ordinary differential equation (ODE)-based framework. This approach is complemented by a Riemannian metric-based regularizer to control the dynamics of weight movement during the fine-tuning process, thereby providing a unified theoretical foundation that connects empirical fine-tuning methods with geometric principles. The authors introduce DELoRA, a method that adapts existing fine-tuning techniques by leveraging differential geometry tools. They demonstrate that DELoRA offers competitive performance across multiple dense prediction tasks—such as semantic segmentation, saliency detection, and surface normal estimation—on the PASCAL-MTL dataset, while achieving reductions in the number of trainable parameters compared to existing approaches.

## Strengths
1. Novel Perspective: The paper introduces a novel geometric perspective on the fine-tuning process. While the theoretical foundation is interesting, it is relatively straightforward and not deeply explored in this work. Nevertheless, the approach offers some insight into the fine-tuning process through the lens of differential geometry.
2. Parameter Efficiency: The proposed method achieves parameter efficiency, using fewer trainable parameters compared to existing methods such as LoRA. This aspect is promising for applications in resource-constrained environments, though the improvement is modest and the evaluation is somewhat limited.
3. Clear Presentation: The paper is clearly written and provides a logical structure, from the introduction of the theoretical framework to the empirical validation. The presentation helps make the core ideas accessible, even if the implementation is not particularly groundbreaking.
4. Experimental Validation: The empirical results, although limited, demonstrate that the proposed method can achieve competitive performance in a multi-task learning setting. The experiments are well-documented and provide a reasonable basis for comparison with existing methods.

## Weaknesses
1. Limited Empirical Demonstration of Generality: The authors only instantiate their geometry-guided fine-tuning (Definition 2.2) in one practical example—its connection to LoRA. This limits the scope of empirical validation and raises questions about the generality and applicability of the proposed framework to other fine-tuning methods. More examples or broader applications would have strengthened the claim of a "unified framework."
2. Clarity of the Unification Claim: The paper claims to provide a "unifying theoretical foundation" for fine-tuning, but this claim is somewhat ambiguous. Although the connection to LoRA is well-explained, it is less clear how broadly applicable the framework is to other fine-tuning techniques or how it generalizes beyond specific cases. Additional discussion on this point, as well as more explicit comparisons to other fine-tuning methods, would be beneficial for clarity.
3. Limited Scope of Evaluation: The experiments are conducted on a single dataset (PASCAL-MTL) with a focus on dense prediction tasks. Evaluating the method on a more diverse set of tasks and datasets could provide a stronger empirical foundation and demonstrate the robustness of the proposed approach.

## Recommendation
While I am not an expert in the fine-tuning of neural networks, I find that the idea is clearly presented, and the geometric perspective on fine-tuning is novel. However, the claim of providing a "unified framework" is less well-substantiated due to limited empirical examples and validation of the approach beyond LoRA. Despite these limitations, given that this paper introduces a promising direction for efficient fine-tuning with reasonable theoretical backing, I believe it passes the bar for a workshop venue. Therefore, I recommend a weak accept.
Suggested Improvements
1. Broader Empirical Validation: To strengthen the claim of generality, it would be helpful if the authors could demonstrate their geometry-guided fine-tuning in more diverse settings beyond the connection to LoRA. Examples could include its application to other fine-tuning strategies or in different model architectures (e.g., CNNs, RNNs).
2. Clarification of Unification Scope: The authors should provide a more detailed discussion of the unification aspect of their framework. Specifically, elaborating on how other fine-tuning methods could be derived or improved using the proposed geometric approach would make the contribution clearer and more compelling.
3. Broader Evaluation: Extending the empirical evaluation to include other datasets or tasks (e.g., text-based tasks or reinforcement learning) could provide additional insights into the method's applicability and limitations. This would strengthen the empirical foundation and support the broader applicability of the approach.

---

### Official Review · Reviewer_mUsX · 2024-10-06
**This paper offers a geometric perspective on LoRA and introduces a new fine-tuning method, DELoRA, by constructing vector fields.**

**Rating:** 6
**Confidence:** 4

**Review:**

##### Summary of Strengths

- The method is innovative, combining the perspective of ODE to analyze low-rank adaptation.
- The proposed geometry-guided fine-tuning is well-structured and supported by experimental validation in the context of multi-task learning.

##### Summary of Weaknesses

- The paper lacks clarity on the specific implementation of multi-task learning, particularly how DELoRA is utilized for fine-tuning across multiple tasks.
- While the Riemannian metric is introduced, it does not seem to be explicitly discussed or utilized in the experiments.
- Lacks sufficient ablation studies and analysis.

---

### Author Response · Authors · 2024-11-01
**Official Response to Reviewers**

# General Revisions and Additional Details
In response to all reviewers, we have made the following additional revisions:
- **Broader Theoretical Discussion:** We expanded the discussion on how DELoRA’s geometric framework and its connection to LoRA, setting a foundation for broader application across different fine-tuning methods.
- **Increased Empirical Analysis:** To strengthen our empirical results, we expanded the discussion on multi-task learning and how DeLoRA is used in conjunction with gradient descent.
- **Citation Updates:** We updated citations, including the recommended references on task arithmetic and other relevant works, to better contextualize our contributions within the broader literature.

In summary, we thank all reviewers for their constructive feedback, which has allowed us to enhance the clarity, depth, and theoretical foundations of our extended abstract. We believe the revisions address the key concerns raised and provide a stronger basis for understanding the impact and generalizability of DELoRA within the geometry-guided fine-tuning paradigm.

---

### Decision · Program_Chairs · 2024-10-10

**Decision:**

Accept

**Comment:**

In light of the positive reviewers' feedback and relevancy of the submission, we are pleased to accept this paper for presentation at UniReps 2024. We kindly ask the authors to incorporate the reviewers' suggestions and feedback in the final camera-ready version of the manuscript.